# GPR183 Is Dispensable for B1 Cell Accumulation and Function, but Affects B2 Cell Abundance, in the Omentum and Peritoneal Cavity

**DOI:** 10.3390/cells11030494

**Published:** 2022-01-31

**Authors:** Line Barington, Liv von Voss Christensen, Kristian Kåber Pedersen, Kristine Niss Arfelt, Martin Roumain, Kristian Høj Reveles Jensen, Viktoria Madeline Skovgaard Kjær, Viktorija Daugvilaite, John F. Kearney, Jan Pravsgaard Christensen, Gertrud Malene Hjortø, Giulio G. Muccioli, Peter Johannes Holst, Mette Marie Rosenkilde

**Affiliations:** 1Laboratory for Molecular Pharmacology, Department of Biomedical Sciences, Faculty of Health and Medical Sciences, University of Copenhagen, 2200 Copenhagen, Denmark; lineb@sund.ku.dk (L.B.); liv.blom@sund.ku.dk (L.v.V.C.); kristian.kp@sund.ku.dk (K.K.P.); krinor@sund.ku.dk (K.N.A.); kristian.reveles@sund.ku.dk (K.H.R.J.); viktoria.kjaer@sund.ku.dk (V.M.S.K.); vdau@sund.ku.dk (V.D.); ghjortoe@sund.ku.dk (G.M.H.); 2Bioanalysis and Pharmacology of Bioactive Lipids Research Group, Louvain Drug Research Institute, Université catholique de Louvain, 1200 Brussels, Belgium; martin.roumain@uclouvain.be (M.R.); giulio.muccioli@uclouvain.be (G.G.M.); 3Division of Developmental and Clinical Immunology, Department of Microbiology, University of Alabama at Birmingham, Birmingham, AL 35294, USA; jfk@uab.edu; 4Infectious Immunology Group, Department of Immunology and Microbiology, Faculty of Health and Medical Sciences, University of Copenhagen, 2200 Copenhagen, Denmark; jpc@sund.ku.dk; 5Experimental Vaccinology Group, Centre for Medical Parasitology, Department of Immunology and Microbiology, Faculty of Health and Medical Sciences, University of Copenhagen, 2200 Copenhagen, Denmark; pjh@inprother.com; 6InProTher ApS, 2200 Copenhagen, Denmark

**Keywords:** GPCR, 7TM receptor, GPR183, EBI2, B1 cell, B-1 cell, oxysterol

## Abstract

B1 cells constitute a specialized subset of B cells, best characterized in mice, which is abundant in body cavities, including the peritoneal cavity. Through natural and antigen-induced antibody production, B1 cells participate in the early defense against bacteria. The G protein-coupled receptor 183 (GPR183), also known as Epstein-Barr virus-induced gene 2 (EBI2), is an oxysterol-activated chemotactic receptor that regulates migration of B cells. We investigated the role of GPR183 in B1 cells in the peritoneal cavity and omentum. B1 cells expressed GPR183 at the mRNA level and migrated towards the GPR183 ligand 7α,25-dihydroxycholesterol (7α,25-OHC). GPR183 knock-out (KO) mice had smaller omenta, but with normal numbers of B1 cells, whereas they had fewer B2 cells in the omentum and peritoneal cavity than wildtype (WT) mice. GPR183 was not responsible for B1 cell accumulation in the omentum in response to i.p. lipopolysaccharide (LPS)-injection, in spite of a massive increase in 7α,25-OHC levels. Lack of GPR183 also did not affect B1a- or B1b cell-specific antibody responses after vaccination. In conclusion, we found that GPR183 is non-essential for the accumulation and function of B1 cells in the omentum and peritoneal cavity, but that it influences the abundance of B2 cells in these compartments.

## 1. Introduction

B1 cells are a subset of B cells defined by their unique phenotypes and functions. They are best characterized in mice, and in mouse spleens they make up only ~2% of B cells, whereas they are much more abundant in the pleural and peritoneal cavities [1,2,3]. The existence of, and identity of, human B1 cells is still controversial [4,5].

B1 cells can be distinguished from conventional (B2) B cells by their surface marker expression profile. B1 cells have a low surface expression of B220 (an isoform of the pan B cell marker CD45R), whereas this marker is highly expressed on B2 cells [6]. Furthermore, the surface marker CD5 groups B1 cells into two subsets; B1a cells, which express CD5, and B1b cells, which do not [7]. 

B1 cells are the primary producers of so-called “natural antibodies”, typically low-affinity IgM antibodies that recognize both self- and foreign antigens and are involved in maintaining tissue homeostasis as well as providing early protection against pathogens [8,9,10,11,12,13]. B1 cells also produce protective antibodies in response to pathogen infection or vaccination, mainly antibodies with specificities for T-independent antigens such as polysaccharides and lipids [13]. 

The peritoneal cavity and omentum are immunological niches containing a large number of B1 cells. The omentum is a visceral adipose tissue encapsulated by mesothelial cells, connecting the spleen, pancreas, stomach, and colon [14]. It contains structured clusters of immune cells, called milky spots (MSs), primarily consisting of B cells, and secondarily T cells, macrophages, dendritic cells, and innate lymphoid cells [14]. The omentum is an immunologically active organ, which, e.g., supports T-dependent antibody responses [15,16]. Furthermore, it was recently shown that the organ plays a major role in the protection against cecal ligation and puncture-induced sepsis [17]. 

The chemokine receptors CXCR5 and CCR7 have been described to be involved (to different degrees) in regulating the migration of B1 cells to and from the peritoneal cavity and omentum [18,19]. However, many aspects of B1 cell migration and function remain underexplored. 

In the spleen, CXCR5 and CCR7 act in concert with the chemotactic G protein-coupled receptor GPR183, also known as Epstein-Barr virus-induced gene 2 (EBI2), to regulate the migration of B cells during a T-dependent antibody response [20,21,22,23]. GPR183 is a G_αi_-coupled receptor responding to a range of oxysterols [24,25,26]. In the spleen, the regulation of GPR183 expression in B cells controls their migration towards its potent oxysterol ligand 7α,25-dihydroxycholesterol (7α,25-OHC), which is found in outer and inter-follicular regions, but not in the follicle centers [27,28]. Consequently, mice with GPR183-deficiency or GPR183-overexpressing B cells develop decreased antibody responses to vaccination with T-dependent antigens [20,21,29].

Thus, we hypothesized that GPR183, being known as an important player in regulating the migration of (B2) B cells in the spleen, is also involved in regulating the migration of B1 cells in the peritoneal cavity and omentum. 

In recent years, GPR183 has been shown to be involved in several local immune responses. Interestingly, GPR183 controls the migration of type 3 innate lymphoid cells, which is a type of cell found in the intestinal mucosa. GPR183-expression in these cells was found to be essential for the formation of lymphoid tissues in the colon [30]. 

Finally, further supporting our hypothesis of a role for GPR183 in B1 cells, we described earlier how B cell-restricted overexpression of GPR183 in mice led to a massive expansion in their B1a cell subset, both in the spleen, blood, and peritoneal cavity [29]. However, in spite of this, these mice had decreased levels of natural antibodies, suggesting that the expression level of GPR183 could be important, not only for the development, but also for the function of B1 cells [29]. 

Here, we investigated the role of GPR183 in B1 cells in the peritoneal cavity and omentum. We found that while B1 cells expressed GPR183 at the mRNA level and migrated towards 7α,25-OHC in vitro, GPR183 was dispensable for the accumulation and function of B1 cells in the peritoneal cavity and omentum. However, GPR183 influenced the abundance of B2 cells in these compartments.

## 2. Materials and Methods

### 2.1. Mice

Homozygous GPR183 knock-out (KO) mice (Gpr183tm1Lex) were originally obtained from Lexicon Pharmaceuticals (The Woodlands, TX, USA). They were back-crossed to a C57BL/6J background by mating with mice obtained from Taconic (Rensselaer, NY, USA). Wild-type (WT) C57BL/6J mice (Taconic) were used as controls. GPR183 KO mice, backcrossed to a C57BL/6JTacBom background and littermate controls were used for the apoptosis experiment. WT C57BL/6JTacBom mice were used for the migration experiment.

All mouse experiments were approved by the Danish Animal Experiments Inspectorate (license numbers 2017-15-0202-00117 and 2018-15-0201-01442) and by the veterinarian unit at the University of Copenhagen. Mice were housed in groups in cages with ad libitum access to food and water. Age-matched female mice (ages are specified in each figure legend) were used for all experiments.

### 2.2. Cell Preparation and Flow Cytometry

Peritoneal lavage was performed by i.p.-injecting 2 mL of saline into euthanized mice, shortly massaging their abdomens to release cells, and then extracting app. 1 mL of lavage fluid. Blood was taken from the tail vein or the jaw into EDTA-containing tubes. Spleens and omenta were mechanically disrupted and passed through a 70 µm nylon mesh. Red blood cells in spleen suspensions and blood were lysed by treating with Gey’s solution. Cells were stained with fluorophore-coupled monoclonal antibodies in PBS buffer containing 10% rat serum, 1% BSA, and 0.1% NaN_3_. 

For all flow cytometry experiments, except the indomethacin challenge, apoptosis detection, and migration experiment, the following markers and monoclonal antibodies were used: CD5 (clone 53-7.3), CD19 (6D5), B220 (RA3-6B2), IgM (RMM-1), and Ki-67 (SolA15). All antibodies were obtained from BioLegend (San Diego, CA, USA). They were directly coupled to phycoerythrin (PE), fluorescein isothiocyanate (FITC), allophycocyanin (APC), pacific blue, or PE-Cy7. For the indomethacin challenge experiment, the markers and fluorophor-coupled monoclonal antibodies used were: CD45-APC/Cy7 (30-F11), CD3-PB (17A2), γδTCR-PE/Cy7 (GL3), CD5-PE (53-7.3), CD4-APC (GK1.5), MHC.II[I-A/I-E]-PE/Cy5 (M5/114.15.2), F4/80-BV605 (BM8), and CD40-FITC (3/23); all from BioLegend, as well as CD19-BV786 (1D3), B220-BV650 (RA3-6B2), Ly6G-BV711 (1A8), and CD8-BUV395 (53-6.7); all from BD Biosciences (Franklin Lakes, NJ, USA). In that particular experiment, all cells were stained with the LIVE/DEAD marker Zombie Aqua (BioLegend cat. 423101) and pre-incubated for 5 min with Tru Stain FcX and anti-mouse CD16/32 antibody (BioLegend, clone 93), before they were incubated with the fluorophor-coupled antibodies. For the apoptosis detection experiment, the markers and fluorophor-coupled monoclonal antibodies used were IgM-PE-Cy7, CD19-PB, CD45R-BV650, and CD5-PE with Annexin V-FITC and PI from FITC Annexin V Apoptosis Detection Kit with PI; all from BioLegend. For the migration experiment, the markers and fluorophor-coupled monoclonal antibodies used were: IgM-PE-Cy7, CD19-PB, CD45R-BV650, and CD5-PE; all from BioLegend.

Cells were incubated with the antibodies for 20 min at 4 °C, washed in PBS buffer with 0.1% NaN3, and fixed in 1% paraformaldehyde. Samples were run on an LSRFortessa flow cytometer (BD Biosciences), and data were analyzed using FlowLogic (Inivai Technologies, Mentone, Australia) or FlowJo software (BD Biosciences). Cell sorting was performed using a FACSAria II cell sorter from BD Biosciences. The purity of the sorted cells was >95%.

### 2.3. RNA Purification and Quantitative Real-Time PCR

RNA was purified from FACS-sorted cell populations using the RNeasy Micro Kit (Qiagen, Venlo, The Netherlands), according to the manufacturer’s instructions. mRNA was then reverse transcribed to cDNA using the SuperScript III Reverse Transcriptase (Thermo Fisher Scientific, Waltham, MA, USA). Quantitative real-time PCR (qPCR) was performed using primers for murine GPR183 (GPR183) (Fw: 5′-CACTGCCGCTGCTCCTCACC-3′, Rv: 5′-GGCAGCACGTAGCCCAGCAG-3′) and for the control gene YWHAZ (Fw: 5′-AGACGGAAGGTGCTGAGAAA-3′, Rv: 5′-GAAGCATTGGGGATCAAGAA-3′). The samples were run on an Mx3000P instrument (Stratagene, La Jolla, CA, USA), using SYBR Premix Ex Taq (Takara, Tokyo, Japan). The Δ-Δ Ct method determined the fold differences in RNA levels between GPR183 and the control gene.

### 2.4. Migration Assay

Mice at age 15–16 weeks were euthanized, and peritoneal lavage was taken as described earlier. Samples were treated with ammonium-chloride-potassium (ACK) and washed in PBS. Cells were re-suspended in assay medium (RPMI1640 with 0.5% BSA) and counted in a microscope using trypan blue. 7α,25-OHC (cat. SML0541, Merck (Sigma-Aldrich), Kenilworth, NJ, USA) and recombinant human CXCL13 (cat. 300-47, Peprotech, Rocky Hill, NJ, USA) were used as ligands, in the concentrations indicated in Figure 1, and placed in the bottom of ChemoTx plates with 5 µm pore size, 30 µL volume, and 3.2 mm diameter (Neuro Probe, Gaithersburg, MD, USA). Then, 150,000 cells in 20 µL medium were added on top of the filter, and the plate was placed in an incubator at 5% CO_2_ for 3 h at 37 °C. After that, migrated cells were counted using flow cytometry. Migrated cells were re-suspended in Fc block, antibodies were added, and cells were prepared and analyzed as earlier described.

### 2.5. Apoptosis Detection

Mice at the age of 11–15 weeks were euthanized, peritoneal lavage was taken, and cells were prepared and analyzed as described under Section 2.2. The experiment was performed according to the manufacturer’s protocol (FITC Annexin V Apoptosis Detection Kit with PI, BioLegend) with an included buffer high in calcium. Antibodies, annexin V, and propidium iodide (PI) was added as an antibody suspension.

### 2.6. Immunizations and ELISAs

In *Streptococcus pneumoniae* vaccination experiments, mice were i.p.-immunized with 10^7^ inactivated *S. pneumoniae* bacteria of the strain R36A (produced in the laboratory of John Kearney) in PBS. Five days later, blood samples were taken, and the serum was analyzed for phosphorylcholine (PC)-specific IgM antibodies by ELISA. Plates were coated at 4 °C overnight with 100 µL of 2 µg/mL PC-BSA (LGC Biosearch Technologies, Hoddesdon, UK) in PBS. Washing was done with PBS, and the plate was blocked with 1% BSA in PBS. Serum from blood samples taken before vaccination was diluted 1:12.5, whereas serum from vaccinated animals was diluted 1:100. The monoclonal anti-PC IgM antibody BH8 (produced by John Kearney) was used to make a standard curve for quantitation of antibody titers. The plate was incubated with serum diluted in PBS for 2 h at 37 °C. For detection, the plate was incubated with a 1:1000 dilution of rat anti-mouse IgM-HRP (clone SB73a, SouthernBiotech, Birmingham, AL, USA) in PBS for one hour at room temperature. The plate was developed with TMB (KEM-EN-TEC Diagnostics, Windsor, CT, USA), color development was stopped with 0.2 M H_2_SO_4_, and the plate was read at 450 nm in an EnVision plate reader (Perkin Elmer, Waltham, MA, USA).

In the 4-hydroxy-3-nitrophenyl-acetyl (NP)-Ficoll vaccination experiment, mice were i.p.-vaccinated with 100 µg NP-Ficoll in PBS. Seven days later, blood samples were taken, and the levels of anti-NP IgM in the serum determined by ELISA. The plate was coated with 100 µL/well of 5 µg/mL NP-BSA, ratio > 20 (LGC Biosearch Technologies) in PBS at 4 °C overnight. The rest of the ELISA was performed like the PC-specific ELISA, except that 1:25 dilutions of serum were used from both unvaccinated and vaccinated animals.

In the adenovirus immunization experiment, mice were i.p.-vaccinated with 10^9^ virus particles of an adenovirus type 5 (Ad5) vector encoding vesicular stomatitis virus (VSV) glycoprotein sequences, called Ad5-VSV. The vector was produced and purified by cesium chloride gradient ultracentrifugation and titrated using the Adeno-X rapid titer kit (Clontech Laboratories, Mountain View, CA, USA). Fourteen days later, serum samples were taken and analyzed for anti-Ad5 IgG antibodies by ELISA. The ELISA plate was coated overnight at 4 °C with 100 µL/well of a solution of 5 × 10^9^ virus particles/mL of heat-inactivated human Ad5 E1 deleted serotype vector in dilution/blocking buffer (PBS with 2% NaCl, 0.5% BSA, 0.05% Tween; used for all incubation steps). Washing was done using PBS with 2% NaCl and 0.1% Tween. The plate was blocked for 1 h at room temperature. Serum samples were diluted 1:25, and from that a 2-fold dilution series was made. The plate was incubated for 1 h at room temperature with 6 different dilutions of each sample as well as of a positive control serum sample, which was known to have a high level of anti-Ad5 IgG, and a negative control serum sample from a mouse that had been vaccinated with an unrelated antigen. For detection, the plate was incubated with a 1:500 dilution of a peroxidase-coupled rabbit anti-mouse IgG1 antibody (SAB3701171, Merck (Sigma-Aldrich)) for 1 h at room temperature. The rest of the procedure was as described above for the PC-specific ELISA.

### 2.7. In Vivo Antagonist Treatment

The GPR183 antagonist NIBR189 (a kind gift from Andreas Sailer, Novartis, Basel, Switzerland) was dissolved in 0.5% carboxymethylcellulose/0.5% Tween80 and given to mice by gavage twice daily at 12 h intervals for one week. The mice received 7.6 mg antagonist/kg bodyweight/dosing. The control group was gavaged with the buffer alone.

### 2.8. Whole-Mount Omentum Microscopy

Whole omenta were fixed for 2 h in 4% paraformaldehyde, washed in PBS and incubated for 1 h at room temperature in PBS containing 5% BSA and 0.2% Triton-X. They were then stained with a 1:100 dilution of rat anti-mouse B220 antibody (clone RA3-6B2, BioLegend) in PBS containing 5% BSA for 4 h at 4 °C. After washing, the omenta were incubated with a 1:500 dilution of donkey anti-rat IgG coupled to Alexa Fluor 594 (cat. no. A-21209, Thermo Fisher Scientific) at 4 °C overnight. After a 20 min incubation with 20 µM Hoechst 33342 (Thermo Fisher Scientific), the omenta were washed and mounted on a slide in ProLong Gold antifade reagent (Invitrogen, Waltham, MA, USA). Images were acquired with an Axio Scan.Z1 slide scanner (Carl Zeiss, Stuttgart, Germany) using the 353/465 nm (Hoechst) and the 577/603 nm (Alexa Fluor 594) channels. The images were analyzed using ZEN Blue software (Carl Zeiss) with the ZEN Intellisis software extension. This software extension was used to predict ‘milky spot’ and ‘non-milky spot’ areas of omenta using artificial intelligence based on previous user-defined examples. 

### 2.9. In Vivo LPS Stimulation

Mice were i.p.-injected with 10 µg of LPS from *E. coli* O111:B4 (Merck (Sigma-Aldrich)) in PBS or with PBS alone. For oxysterol measurements (see below), mice were euthanized after 6 h. In the experiments investigating the migration of B cells to the omentum in response to LPS challenge, the mice were euthanized after 18 h. 

### 2.10. Oxysterol Measurements

Snap-frozen peritoneal lavage samples and whole omenta were sent to the laboratory of Giulio Muccioli, where their oxysterol levels were measured. Standards (oxysterols and deuterated oxysterols) were bought from Avanti Polar Lipids (Birmingham, AL, USA). Free oxysterols were extracted, pre-purified and quantified using a validated high-performance liquid chromatography-mass spectrometry (HPLC-MS) method, as described previously [31,32]. Briefly, deuterated standards were added to the peritoneal lavage and to homogenized omenta and the oxysterols extracted using a dichloromethane, methanol, and water mixture (in the presence of butylated hydroxytoluene and ethylenediaminetetraacetic acid to minimize oxidation). Following pre-purification using a silica column, the oxysterol fraction was analyzed by HPLC-MS using an LTQ-Orbitrap mass spectrometer. Xcalibur (from Thermo Fisher Scientific) was used for both data acquisition and processing. Oxysterols were quantified by the isotope dilution method, using as internal standards d_7_-4β-OHC (for the oxysterols oxidized on the sterol backbone), and d_7_-24-OHC (for those oxidized on the lateral side chain). The calibration curves were built in the same conditions [33]. 

### 2.11. In Vivo Indomethacin Challenge

One week before indomethacin treatment, the mice were randomly grouped and placed in their respective cages to acclimatize them to their new housing. The indomethacin was administered through drinking water for 1 week total at a concentration of 10 mg/kg/day. To prepare the indomethacin-water, a stock solution of 30–60 mg indomethacin (Merck (Sigma-Aldrich), CAS: 53-86-1) in 2 mL 96% ethanol was prepared and added to 500–1000 mL of tap water. The final concentration of indomethacin in the indomethacin-water was based on the mean weight of all mice as well as a daily intake of 5 mL of water per mouse. The indomethacin-water was changed every second day during the experiment. At the end of the experiment, all mice were euthanized and their spleens were weighed. Immune cell subsets were analyzed by flow cytometry as described above.

### 2.12. Statistical Analysis

Where nothing else is mentioned, data was processed using Excel (Microsoft Corporation, Redmond, WA, USA) or GraphPad Prism (GraphPad Software, San Diego, CA, USA). Determination of statistical significance was done using the tests described in each figure legend. A *p*-value of <0.05 was considered statistically significant. 

## 3. Results

### 3.1. B1 Cells of the Peritoneal Cavity Express GPR183 and Migrate in Response to 7α,25-OHC

To determine if B1 cells express GPR183, we FACS-sorted cells from the peritoneal cavity of C57BL/6 mice into B1a, B1b, and B2 cell subsets (Figure 1A). We first gated out on lymphocytes (not shown) and sub-gated for B cells (IgM^+^CD19^+^), see Figure 1A—left panel. Then, we sub-gated the B cells for B1a (B220^low/÷^CD5^+^), B1b (B220^low/÷^CD5^÷^) and B2 (B220^hi^CD5^÷^) cells, see Figure 1A—right panel. CD5 in itself is not a B1 cell-specific marker, as it is also expressed on anergic B cells in mice [34]. However, in combination with B220, it can be used to distinguish between the two B1 cell subsets. qPCR revealed that all three B cell subsets expressed GPR183 at the mRNA level, but B2 cells had significantly higher relative amounts of GPR183 mRNA than B1a and B1b cells (Figure 1B). We did not have a suitable antibody to determine GPR183 protein levels. However, we tested the migration of B1 and B2 cells in response to GPR183′s potent oxysterol ligand 7α,25-OHC in an in vitro migration system. The migration of C57BL/6 mouse peritoneal cavity cells in response to 7α,25-OHC was assessed by identifying and quantifying the migrated cells by flow cytometry. Peritoneal cavity cells in general and B1 cells in particular migrated in response to 7α,25-OHC in a dose-dependent manner (Figure 1C). By assessing whether the B1 cell frequencies out of total B cells in the cell samples changed after migration, we were able to test if B1 cells migrated to a different extent than B2 cells. We found that B1 cell frequencies were not significantly different after migration in response to 7α,25-OHC, suggesting that B1 (B1a + B1b cells; the low numbers of B1b cells in this experiment made it too uncertain to conclude anything based on this specific population) and B2 cells migrated to a similar extent in response to this ligand. In contrast, there was a significantly increased frequency of B1 cells after migration in response to the CXCR5 ligand CXCL13, suggesting that B1 cells migrate to a higher degree than B2 cells in response to CXCL13 (Figure 1D), as proposed in a previous study [18].

### 3.2. GPR183 Impacts B1 and B2 Cell Numbers in the Spleen, Blood, Peritoneal Cavity, and Omentum

To assess whether GPR183 expression impacts B1 cell numbers, we measured the numbers of B1 cells in different tissues by flow cytometry in GPR183 KO compared to WT mice. GPR183 KO mice had significantly more B1a cells, but not B2 cells, in the spleen (Figure 2A). Similarly, in the blood, GPR183 KO mice had comparable frequencies of B2 cells, but higher frequencies of B1a cells among total lymphocytes (Figure 2B). Due to the low numbers of B1b cells in the spleen and blood, the cell frequencies could not be determined with certainty with our gating strategy (Figure 1A).

In contrast, in the peritoneal cavity and the omentum, GPR183 KO mice had significantly fewer B2 cells and similar numbers of B1a and B1b cells (Figure 2C,D). To determine if the lack of GPR183 signaling in the GPR183 KO mice was responsible for the smaller B2 cell population in the peritoneal cavity, we used NIBR189, a specific GPR183 antagonist [35]. GPR183 KO mice had lower (although not significantly in this experiment) frequencies of B2 cells of total peritoneal cells than WT mice (Figure 3A), supporting the findings in Figure 2C.

Interestingly, like GPR183 KO mice, antagonist-treated WT mice had lower (though still not significantly) frequencies of B2 cells than WT mice (Figure 3A). GPR183 KO mice also had higher frequencies of B1b cells in this experiment (Figure 3A). A previous study suggested that GPR183 positively regulates B cell proliferation [36]. We, therefore, considered if decreased B cell proliferation could explain the lower numbers of B2 cells in GPR183 KO and antagonist-treated mice. However, the frequencies of cells that expressed the proliferation marker Ki-67 did not match the overall cell frequencies (Figure 3B compared to Figure 3A). Thus, it is improbable that GPR183 controls the frequencies of peritoneal B cell subsets by affecting their proliferation rates.

Another possible explanation for the lower numbers of B2 cells in GPR183 KO mice could be increased apoptosis of these cells. To test this hypothesis, we stained the peritoneal cells from GPR183 KO and WT mice with Annexin V in combination with PI and analyzed the percentages of alive, non-apoptotic B cells by flow cytometry. Interestingly, we found a slightly lower percentage of alive B2 cells from GPR183 KO compared to WT mice (Figure 3C). There was no significant difference for B1 cells (B1a + B1b; combined due to low numbers of B1b cells). To investigate potential functional consequences of the lower number of B2 cells in the peritoneal cavity of GPR183 KO mice, we tested IgG responses to i.p. vaccination with adenovirus Ad5. GPR183 KO and WT mice mounted similar responses 14 days after vaccination (Figure 3D). This suggested that the fewer B2 cells does not result in impaired B2 cell-specific antibody responses.

### 3.3. GPR183 Influences the Size, but Not the Overall Structure, of the Omentum

Since GPR183 KO mice had fewer B2 cells in the omentum (Figure 2D), we wanted to know if the lower abundance affects their omentum structure. For this purpose, whole omenta from GPR183 KO and WT mice were stained with a B cell-specific antibody to visualize milky spots (MSs). MSs were present in omenta from both genotypes of mice (Figure 4A and Appendix AA), however, omenta from GPR183 KO mice were significantly smaller (Figure 4B). To quantify numbers and sizes of MSs, the omentum images were analyzed using artificial intelligence software (Appendix AB). Although not significant, GPR183 KO mice tended to have a smaller total MS area (Figure 4C), due to a tendency to have fewer MSs (Appendix AA) and not due to differences in MS sizes (Appendix AB). However, when we normalized the total MS area and the total MS number to the omentum sizes, the tendencies disappeared (Appendix AC,D). Thus, the main difference between GPR183 KO and WT omenta was their sizes.

### 3.4. GPR183 Is Dispensable for LPS-Induced Migration of Peritoneal Cavity B1 Cells to the Omentum

Although GPR183 KO mice had normal B1 cell numbers in the peritoneal cavity and omentum at steady-state (Figure 2C,D), GPR183 could potentially still play a role in B1 cell migration in response to a non-homeostatic stimulus, e.g., bacteria or bacterial products in the peritoneal cavity. Since several studies show that LPS exposure up-regulates GPR183 expression and stimulates the production of its main ligand 7α,25-OHC [33,37,38], we speculated that LPS-induced production of one or more of GPR183′s ligands in the peritoneal cavity and/or the omentum could be involved in controlling B1 cell migration in this niche. I.p. LPS injection led to a massive increase in 7α,25-OHC amounts in the peritoneal cavity (Figure 5A) and the omentum (Figure 5B). It also resulted in markedly increased amounts of 25-OHC, a less potent GPR183 ligand, in both compartments (Appendix AA,B). Previous studies have shown that in response to i.p. LPS injection, B1 cells migrate out of the peritoneal cavity through the omentum to the spleen and the gut, where they differentiate into antibody-secreting cells [13,39,40,41]. To test if GPR183 is involved in the LPS-induced migration of B1 cells to the omentum, we assessed the accumulation of B cells in the omentum after 18 h. B2 cells did not accumulate in the omentum in response to LPS injection (Figure 5C). In contrast, B1 cells accumulated in the omentum in WT mice. Thus, there were significantly more B1b and a (non-significant) tendency to more B1a cells in the omenta of LPS-injected than in PBS-injected mice (Figure 5C). However, B1 cells accumulated similarly in GPR183 KO mice (Figure 5C), suggesting that GPR183 is not important for the LPS-induced migration of B1 cells out of the peritoneal cavity.

### 3.5. GPR183 Affects the Accumulation of B1 Cells in the Peritoneal Cavity after Indomethacin Treatment

To further pursue whether GPR183 plays a role in the response of peritoneal B1 cells to bacterial or inflammatory stimuli in a different model, we treated GPR183 KO and WT mice with indomethacin, a nonsteroidal anti-inflammatory drug (NSAID) with the ability to increase the permeability of the intestinal epithelial barrier and to cause intestinal inflammation [42,43,44]. Indomethacin has also been reported to cause bacterial translocation out of the intestine in rats [43,45]. Indomethacin was administered in the drinking water over 7 days and the experiment terminated on day 8. GPR183 KO and WT mice reacted in a similar way to the challenge: both lost weight equally during the experiment (Figure 6A), and their spleens were equally heavier than for a normal female C57BL/6 mouse (i.e., <0.2 g [29], Figure 6B). We then assessed the B cell populations in the peritoneal cavity after indomethacin treatment. Compared to the homeostatic situation (Figure 2C), the numbers of both B1 and B2 cells in WT mice were markedly higher after indomethacin treatment (Figure 6C). Interestingly, the numbers of both B1 and B2 cells were also markedly higher in GPR183 KO mice after indomethacin treatment than at homeostasis (Figure 6C). In fact, the numbers of B1a cells were much higher in GPR183 KO than in WT mice. Furthermore, there were equal numbers of B2 cells in the two mouse genotypes after indomethacin treatment, the latter in contrast to the B2 cell deficiency observed in GPR183 KO mice at steady state (Figure 6C compared to Figure 2C). This could suggest that GPR183 may play a role in B2 cell recruitment at homeostasis, but not during inflammation. We found no differences in the abundance of other immune cell subsets, i.e., CD4^+^, CD8^+^, and γδ-T cells, large peritoneal macrophages or small peritoneal macrophages, after indomethacin treatment (Appendix A).

### 3.6. GPR183 Is Not Important for B1 Cell Antibody Responses

Although GPR183 KO mice had unaltered B1 cell quantities in the peritoneal cavity and omentum (Figure 2C,D), and GPR183 was not required for B1 cell migration to the omentum in response to LPS stimulation (Figure 5C), the receptor could still play a role in B1 cell-specific antibody production, as has previously been reported for B2 cell responses in the spleen [20,21,29]. It is well-characterized that B1a cells produce IgM antibodies specific for phosphorylcholine (PC) [46]. These antibodies are present as natural antibodies in the serum of unvaccinated mice, but are highly induced by vaccination with PC-expressing *S. pneumoniae* bacteria [47]. Marginal zone B cells are also able to produce anti-PC IgM antibodies. However, when mice are vaccinated i.p. with a low dose of bacteria, the response is mediated almost exclusively by B1a cells in the peritoneal cavity [10]. To test if GPR183 is important for B1a cell responses in the peritoneal cavity, we determined PC-binding IgM serum concentrations before and after i.p.-vaccination with a low dose of *S. pneumoniae* bacteria. GPR183 KO and WT mice produced equal amounts of PC-specific IgM before and after the vaccination (Figure 7A), suggesting that GPR183 is not essential for the B1a-mediated antibody response against PC on *S. pneumoniae*. B1b cells are responsible for the IgM response to the experimental immunogen NP-Ficoll [48]. GPR183 KO and WT mice responded similarly to vaccination with NP-Ficoll (Figure 7B). This demonstrated that GPR183 is not essential for the antibody response meditated by B1a cells against PC on *S. pneumoniae* or by B1b cells against NP-Ficoll.

## 4. Discussion

We investigated the role of GPR183 in B1 cells of the omentum and peritoneal cavity. Our results showed that GPR183 was not essential for the migration or function of B1 cells in these compartments. Thus, although B1 cells from the peritoneal cavity expressed GPR183 (on the RNA level; Figure 1B) and migrated in response to GPR183′s main ligand, 7α,25-OHC (Figure 1C), we found a normal number of B1 cells in the peritoneal cavity (Figure 2C, confirming a previous study [21]) and in the omentum (Figure 2D) in GPR183 KO mice. A study suggests that the chemokine receptor CXCR5 is the main receptor responsible for the migration of B1 cells to the peritoneal cavity and omentum, as it was shown that mice lacking CXCL13, the only ligand for CXCR5, had markedly decreased numbers of B cells, both B1 and B2, in these compartments (there was a 50-fold reduction in B1a cells). Furthermore, CXCL13 KO mice had a markedly lower accumulation of WT B1 and B2 cells in the peritoneal cavity after intra-venous adoptive transfer [18]. The chemokine receptor CCR7 has also been described to be involved in B1 cell migration [19]. Interestingly, CXCR5 and CCR7 are the two main regulators of immune cells, including B cell, migration in the spleen, where GPR183 plays a more subtle role (reviewed in [28]). Thus, whereas mice lacking CXCR5 or CCR7 have abnormal spleen structures, spleens of GPR183 KO mice appear morphologically normal [20,21,49,50]. Furthermore, GPR183′s role has in some cases proved to be sub-dominant to those of other receptors, e.g., its role in controlling the numbers of marginal zone (MZ) B cells in the spleen. Whereas GPR183 KO and CCR7 + CXCR5 double-KO mice had similar numbers of MZ B cells in the spleen as WT mice, CCR7 + CXCR5 + GPR183 triple-KO mice had significantly more MZ B cells than WT mice [51]. Thus, it is possible that a role for GPR183 in B1 cell migration and function could become apparent in studies using, e.g., multiple KO mouse models.

One of the largest differences in B cell numbers in our study was in the B2 cell subset in the omentum and peritoneal cavity, where WT mice had 2–3 times as many B2 cells as GPR183 KO mice (Figure 2C,D). This could suggest that GPR183 plays a role in B2 cell recruitment to the omentum and peritoneal cavity. Our finding that peritoneal B2 cells express higher levels of GPR183 (at least on the RNA level; Figure 1B) than peritoneal B1 cells supports a specific role of GPR183 in B2 cell recruitment to the peritoneal cavity. However, that peritoneal B1 and B2 cells migrated equally well in response to 7α,25-OHC ex vivo (Figure 1D) could speak against this. As discussed above, the CXCR5 ligand CXCL13 was found to recruit B1 and B2 cells to the omentum and peritoneal cavity. However, B1 and B2 cells were not equally sensitive to peritoneal CXCL13, since CXCL13-deficient mice had a 15-fold reduction in B2 cell recruitment but a 54-fold reduction in B1 cell recruitment [18]. This suggests that CXCR5 is not the only receptor responsible for B2 cell recruitment to the peritoneal cavity. Hypothetically, GPR183 could be another receptor involved. The observation that treatment of WT mice with a GPR183 antagonist decreased the peritoneal B2 cell numbers to that of GPR183 KO mice (Figure 3A) strongly indicates that lack of GPR183 signaling directly causes the lower peritoneal and omental B2 cell numbers. Our data suggest that enhancement of B cell proliferation by GPR183 (which a study reported [36]) likely did not explain the lower B2 cell numbers in GPR183 KO and antagonist-treated mice (Figure 3B). However, an anti-apoptotic effect of GPR183, resulting in an increased apoptosis of B2 cells in GPR183 KO mice, is consistent with our data (Figure 3C), and with a study showing that B cells from a B cell-restricted GPR183 overexpression mouse model had decreased apoptosis in response to etoposide treatment ex vivo [29].

Although omenta of GPR183 KO mice had markedly decreased numbers of B2 cells, we did not see any gross differences in their structure (MS frequencies or sizes, Appendix AB,C). However, GPR183 KO omenta were significantly smaller than WT (Figure 4B). How GPR183 would affect the omentum size is not so clear. Numbers of both immune and non-immune cells were lower in GPR183 KO mice than in WT mice (data not shown). However, preliminary data suggest that adipocytes express GPR183 (data not shown). Therefore, GPR183 KO mice may develop smaller omenta due to defects in both leukocytes and adipocytes.

As described earlier, i.p. LPS injection induces migration of B1 cells to the omentum [40,41]. Using this model, we found that GPR183 was not responsible for this migratory response (Figure 5C). However, i.p. LPS injection dramatically induced the levels of GPR183′s potent ligand 7α,25-OHC, both in the peritoneal cavity and the omentum (Figure 5A,B), which is consistent with what has been observed in the liver and brain [33]. GPR183 signaling has been found to decrease the levels of LPS-induced pro-inflammatory cytokines, both in vitro and in vivo [33,52,53,54]. Thus, a dampening of the immune response may be a GPR183-dependent effect of LPS-induced 7α,25-OHC production in the peritoneal cavity and omentum, rather than an induction of B1 cell migration.

We also assessed the role of GPR183 in B1 cells in an indomethacin challenge model, where all mice had splenomegaly (Figure 6B), suggesting that the indomethacin caused systemic inflammation in both genotypes of mice. Again, we found GPR183 not to be essential for the accumulation of B1 cells in response to the challenge. In fact, we found a higher number of B1 cells (significant for B1a cells) and an equal number of B2 cells in the peritoneal cavity of GPR183 KO compared to WT mice after one week of indomethacin treatment (Figure 6C). The equal B2 cell numbers reflect a large increase from the low number of B2 cells in GPR183 KO mice at steady state (Figure 2C), which may explain why the two genotypes of mice had similar B2-dependent IgG responses to vaccination with adenovirus (Figure 3D). Injection of the virus and the subsequent inflammatory responses may have resulted in an adjustment of the normally low cell numbers up to the level of WT mice. The abundance of B1 and B2 cells in GPR183 KO mice after indomethacin treatment may suggest that GPR183 plays an inhibitory role in gut inflammatory responses by limiting B cell availability. Alternatively, it may simply imply that GPR183 is not essential for recruiting B cells to the peritoneal cavity in response to inflammatory stimuli, and that recruitment by other receptors may be sufficient to compensate for the few B2 cells in the mouse model at steady state. Furthermore, the abundance of B1 cells may somehow reflect the high numbers of B1 cells in the spleen and blood at homeostasis (Figure 2A,B).

In conclusion, we show that GPR183 is not essential for the migration or function of B1 cells in the peritoneal cavity and omentum. It remains a possibility that the receptor plays a sub-dominant role to that of other chemotactic receptors, such as CXCR5 and CCR7, similarly to the role it plays in regulating immune cell migration in the spleen.

## Figures and Tables

**Figure 1 cells-11-00494-f001:**
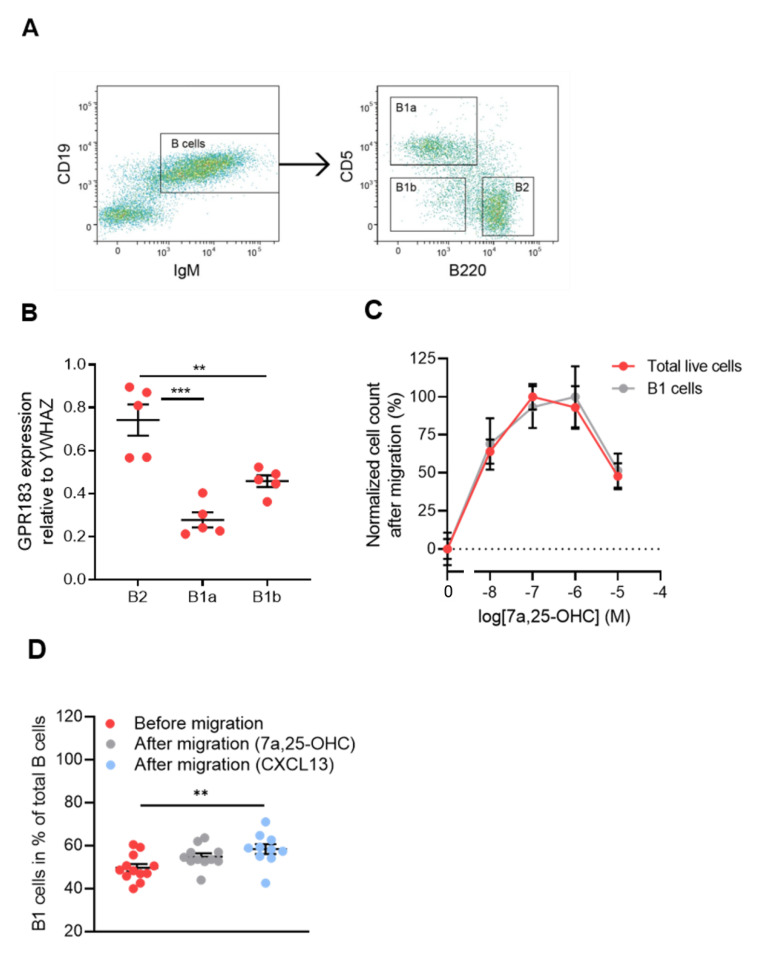
GPR183 expression in and 7α,25-OHC-directed migration of peritoneal cavity B cell subsets; (**A**) gating scheme used to sort peritoneal cavity cells into B1a (B220^low/÷^CD5^+^), B1b (B220 ^low/÷^CD5^÷^), and B2 (B220^+^CD5^÷^) cell subsets. Cells were gated on lymphocytes; (**B**) qPCR analysis of GPR183 mRNA levels relative to the control gene YWHAZ in the indicated B cell subsets sorted from peritoneal cavity cells by FACS, as shown in (**A**). Error bars represent mean ± SEM of data from 5 15–16 weeks old WT mice; (**C**) quantification of cells that migrated in vitro in response to varying concentrations of 7α,25-OHC, as indicated. Identification and quantification of the cells was performed after migration by flow cytometry. Cell counts were normalized to the average highest and lowest values in each dataset; (**D**) frequencies of B1 cells before migration and after migration in response to 10^−7^ M 7α,25-OHC or 10^−7^ M CXCL13. Error bars in (**C**,**D**) represent mean ± SEM of data from 10–12 WT mice at 15–16 weeks of age. ** *p* < 0.01 and *** *p* < 0.001 by one-way ANOVA.

**Figure 2 cells-11-00494-f002:**
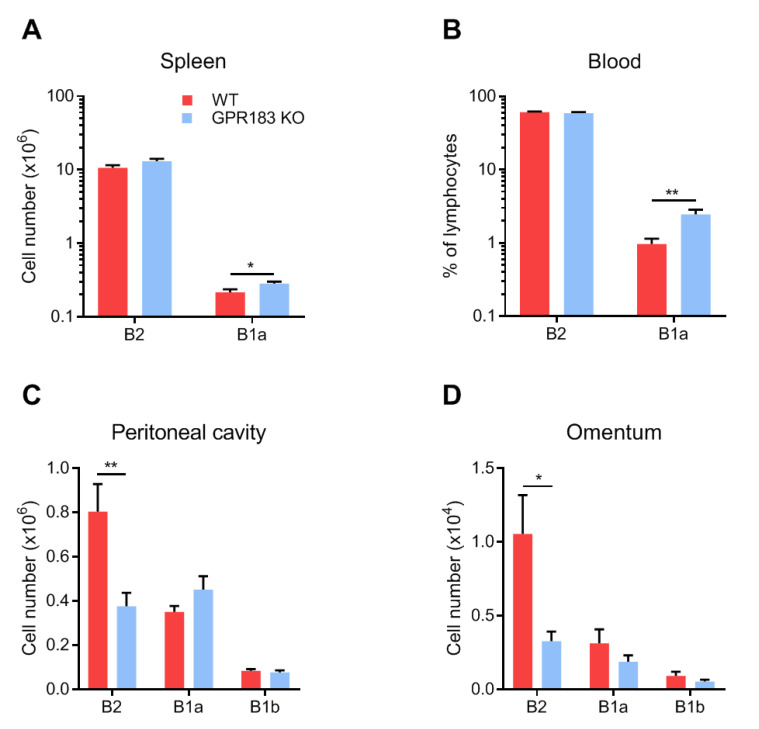
Numbers of B cells in different body compartments of GPR183 KO mice. Numbers or frequencies of cells of the indicated B cell subsets, as determined by flow cytometry. Quantification of cells from; (**A**) spleen; (**B**) blood; (**C**) peritoneal cavity; and (**D**) omentum from 9–19 GPR183 KO and WT mice, which were 11–37 weeks of age. In (**C**) and (**D**), data were pooled from four independent experiments. Error bars represent mean ± SEM. * *p* < 0.05 and ** *p* < 0.01 by unpaired Student’s *t*-test.

**Figure 3 cells-11-00494-f003:**
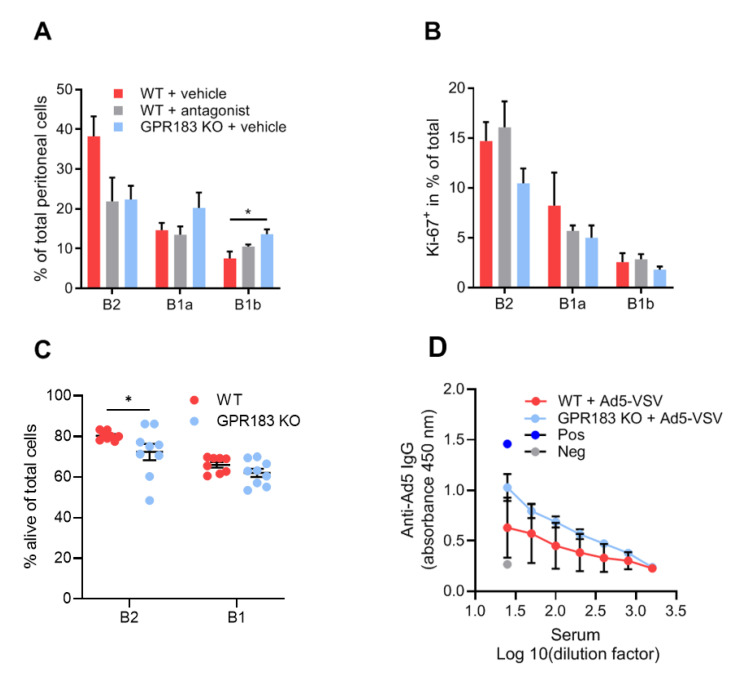
Effects of lack of GPR183 expression or GPR183 signaling on peritoneal cavity B cell frequencies, proliferation, apoptosis, and function; (**A**) and (**B**) mice were given the GPR183 antagonist NIBR189, or vehicle alone, by gavage twice daily for one week. After one week, their peritoneal cavity cells were analyzed by flow cytometry; (**A**) percentages of the indicated B cell subsets in the peritoneal cavity of antagonist-treated WT mice and vehicle-treated WT and GPR183 KO mice; (**B**) percentages of cells of the indicated B cell subsets which are positive for the Ki-67 proliferation marker in antagonist- and vehicle-treated WT mice and vehicle-treated GPR183 KO mice. Error bars represent mean ± SEM of data from 4–5 mice of 17–39 weeks of age. * *p* < 0.05 by one-way ANOVA; (**C**) percentages of alive cells in B cell subsets from the peritoneal cavity of GPR183 KO and WT mice as determined by flow cytometry (living cells do not stain with Annexin V or PI). Error bars represent mean ± SEM of data from 8–9 mice of 11–15 weeks of age. * *p* < 0.05 by one-tailed unpaired *t*-test with Welch’s correction; (**D**) mice were vaccinated with 10^9^ Ad5-VSV virus particles i.p., and 14 days later serum samples were taken and analyzed for Ad5-specific IgG by ELISA. Error bars represent mean ± SEM of data from 4 mice, aged 21–22 weeks. Pos = positive control serum sample known to have anti-Ad5 IgG antibodies. Neg = negative control serum sample from mouse vaccinated with unrelated antigen.

**Figure 4 cells-11-00494-f004:**
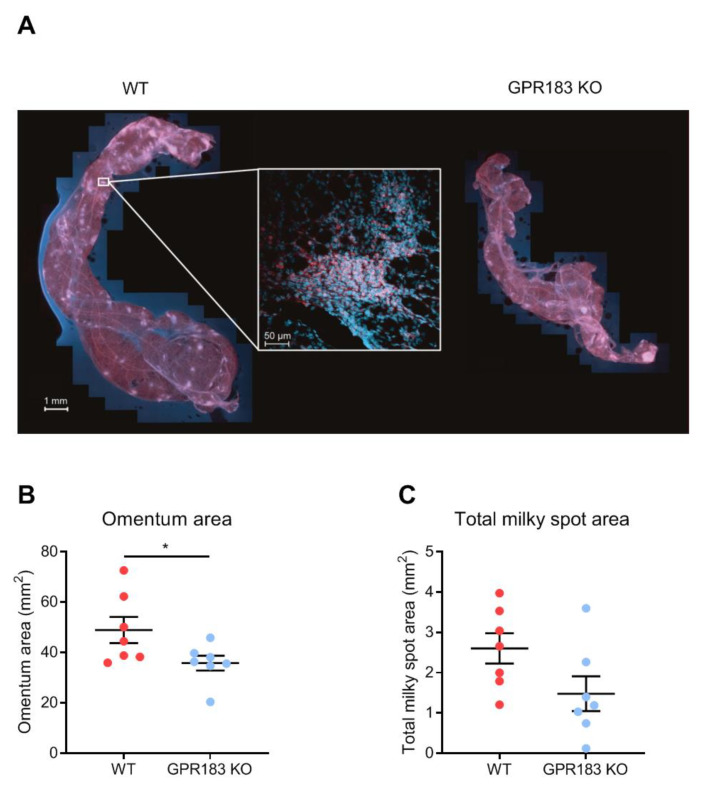
Omentum and milky spots in GPR183 KO mice; (**A**) whole omenta from WT (**left**) and GPR183 KO (**right**) mice stained with an anti-B220 antibody (red) and a Hoechst nuclear stain (blue) and imaged by fluorescence confocal microscopy. The insert shows a magnification of a single milky spot; (**B**) quantification of the total surface area of omenta from 7 10–14 weeks old WT and GPR183 KO mice; (**C**) quantification of the total area per omentum that constitutes milky spots, in the same omenta as in (**B**). The data were pooled from two independent experiments. Error bars represent mean ± SEM. * *p* < 0.05 by unpaired Student’s *t*-test.

**Figure 5 cells-11-00494-f005:**
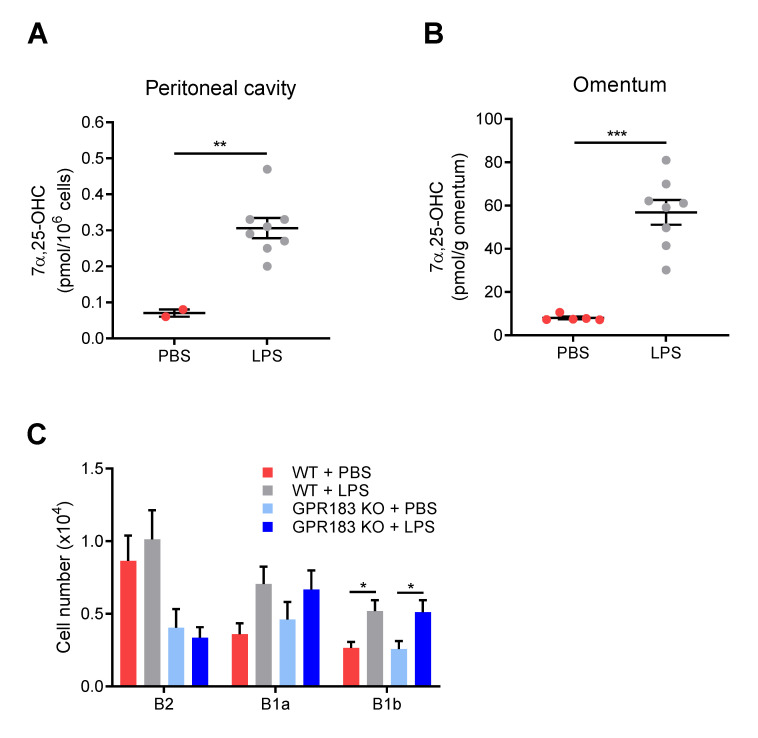
Effects of intra-peritoneal (i.p.) LPS injection on oxysterol levels, and on B cell migration in GPR183 KO mice; (**A**) and (**B**) mice were i.p.-injected with 10 µg LPS in PBS or PBS alone, and tissues were analyzed after 6 h. The graphs show normalized amounts of the oxysterol 7α,25-OHC in total peritoneal lavage (**A**) and omentum (**B**) from 2–8 15 weeks old WT mice, as measured by mass spectrometry. There were 8 mice per group, but in some samples from the PBS-treated group, the amounts were below the detection limit, and these samples were left out. Error bars represent mean ± SEM. ** *p* < 0.01 and *** *p* < 0.001 by unpaired Student’s *t*-test; (**C**) GPR183 KO and WT mice were i.p.-injected with 10 µg LPS in PBS or PBS alone. The graph shows the total number of cells of the indicated B cell subsets in the omentum after 18 h, as determined by flow cytometry. Error bars represent mean ± SEM of data from 17–20 mice of 9–18 weeks of age. The data were pooled from two independent experiments. * *p* < 0.05 by one-way ANOVA.

**Figure 6 cells-11-00494-f006:**
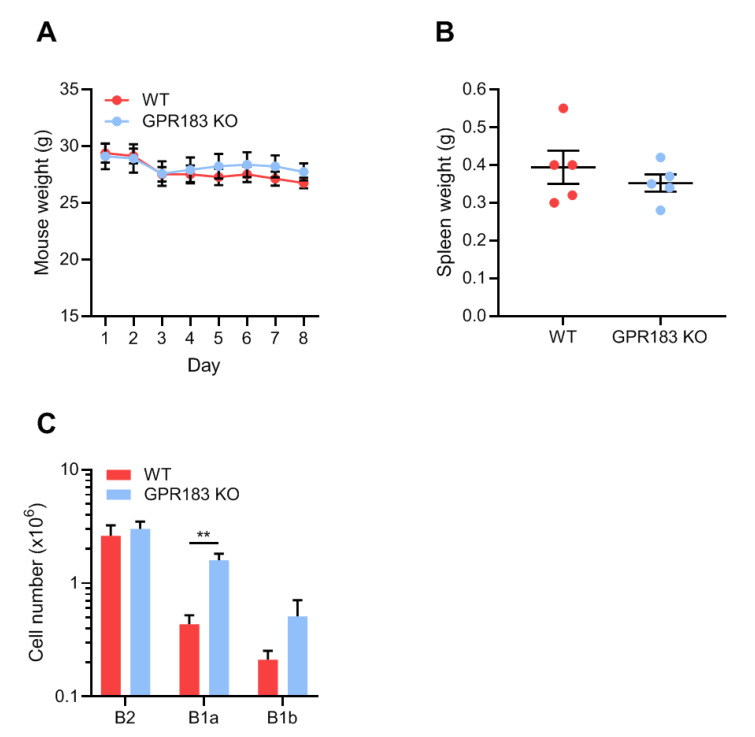
Effects of indomethacin treatment on B cell accumulation in the peritoneal cavity of GPR183 KO mice. GPR183 KO and WT mice were given 10 mg/kg/day indomethacin in their drinking water for 7 days. The experiment was terminated on day 8 and different tissues were analyzed; (**A**) mouse weights during the course of the experiment, where day 1 is the first day of indomethacin treatment and day 8 is the day the experiment was terminated. One GPR183 KO mouse became too ill and was euthanized after day 4 and data for this mouse is included up until and including day 4; (**B**) spleen weights on day 8; (**C**) quantification of cells from different B cell subsets in the peritoneal cavity, as determined by flow cytometry. Error bars represent mean ± SEM of data from 5–6 mice, aged 28–53 weeks. ** *p* < 0.01 by unpaired Student’s *t*-test.

**Figure 7 cells-11-00494-f007:**
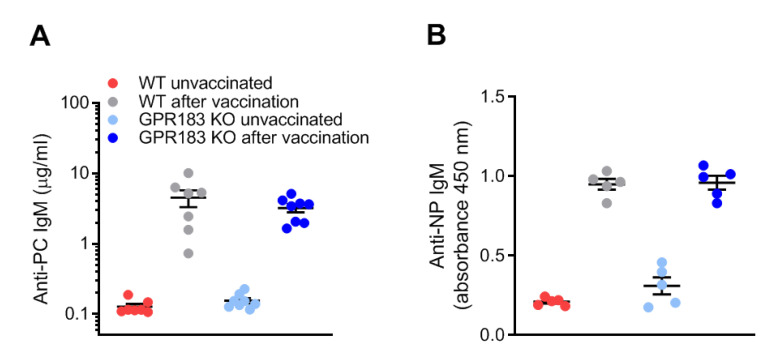
B1 cell antibody responses in GPR183 KO mice; (**A**) GPR183 KO and WT mice were vaccinated i.p. with 10^7^ inactivated *S. pneumoniae* bacteria. Serum phosphorylcholine (PC)-specific IgM levels were determined before and 5 days after the vaccination by ELISA; (**B**) GPR183 KO and WT mice were vaccinated i.p. with 100 µg NP-Ficoll. Serum anti-NP IgM levels were determined before and 7 days after the vaccination by ELISA. Error bars represent mean ± SEM of data from 5–8 mice, aged 13–20 weeks.

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
