# Peer review of "GPR183 Is Dispensable for B1 Cell Accumulation and Function, but Affects B2 Cell Abundance, in the Omentum and Peritoneal Cavity"

_cells, 2022, doi:10.3390/cells11030494_

Round 1

Reviewer 1 Report

This is a well-written manuscript on an interesting subject. However, it could benefit from some more explanations. For instance, the FACS markers could be briefly commented with regard to their specificity. Also, the companies should be complemented by their locations. 

Author Response

Dear Reviewer,

Thank you for your work.

We have the following responses to your comments:

This is a well-written manuscript on an interesting subject. However, it could benefit from some more explanations. For instance, the FACS markers could be briefly commented with regard to their specificity. Also, the companies should be complemented by their locations. 

We have edited the manuscript text (please see attached document with changes in track-changes) and added some more explanations about our gating strategy (in the beginning of the Results section) and about the specificity of CD5 as a cell marker. We have also added a bit of explanation about the B220 marker in the beginning of the Introduction section. We hope that you are satisfied with these additions.

Additionally, we have added the locations of all the companies.

Best Regards,

Line Barington, on behalf of Mette Rosenkilde

Reviewer 2 Report

Barington et al. present a series of pharmacological and in vivo studies in order to investigate the role of GPR138 in the functions of B1 cells. The authors have found that the GPR183 has no too much role in the function of B1 cells in the omentum and peritoneal activity. The choice of the animal model and the biochemical methods is appropriate. The study is well designed and the results support the conclusions of the authors. I have several criticisms and questions to make:

  1. Generally, since the role of GPR183 is not remarkable in B1 cell functions, I would highlight the B2 cell related results in the title, abstract and other places in the text.
  2. I do not understand the Figure 1C. The average of B1+B2 should be 100%, or I miss something here?
  3. It is stated in the abstract that “B1 cells expressed GPR183 at the mRNA level and migrated towards the GPR183 ligand 7α,25-dihydroxycholesterol (7α,25-OHC).”. It is shown in the Figure 1C that the B1 and B2 frequencies are not changed before and after migration but I cannot see that how the actual amplitude of the migration is changed in response to 7α,25-OHC (control vs 7α,25-OHC).  

Author Response

Dear Reviewer,

Thank you for your work.

We have the following responses to your comments:

Barington et al. present a series of pharmacological and in vivo studies in order to investigate the role of GPR138 in the functions of B1 cells. The authors have found that the GPR183 has no too much role in the function of B1 cells in the omentum and peritoneal activity. The choice of the animal model and the biochemical methods is appropriate. The study is well designed and the results support the conclusions of the authors. I have several criticisms and questions to make:

  1. Generally, since the role of GPR183 is not remarkable in B1 cell functions, I would highlight the B2 cell related results in the title, abstract and other places in the text.

We have edited the manuscript text (please see attached document with changes in track-changes) and highlighted the B2 cell-related results in the title, the abstract, and in the end of the Introduction section. 

  1. I do not understand the Figure 1C. The average of B1+B2 should be 100%, or I miss something here?

  1. It is stated in the abstract that “B1 cells expressed GPR183 at the mRNA level and migrated towards the GPR183 ligand 7α,25-dihydroxycholesterol (7α,25-OHC).”. It is shown in the Figure 1C that the B1 and B2 frequencies are not changed before and after migration but I cannot see that how the actual amplitude of the migration is changed in response to 7α,25-OHC (control vs 7α,25-OHC).  

In response to points 2 and 3: We have changed Figure 1 by adding a new Fig. 1C which shows that B1 cells migrate in a dose-dependent manner to 7a,25-OHC. Furthermore, we have added a new Fig. 1D, which shows the frequencies of B1 cells before and after migration to 7a,25-OHC (as in previous Fig. 1C) as well as after migration to CXCL13. We have taken the B2 cell frequencies out, since it is not necessary to show both. However, you are right in that the B1 + B2 frequencies did not add fully up to 100% - that was due to our gating strategy, where we did not gate out 100% of the B cells, because some fell outside the gates. However, as we used the same gates before and after migration, we do not think it is a problem.  

Additionally, we have improved the Introduction. We hope that you agree.

Best Regards,

Line Barington, on behalf of Mette Rosenkilde

Round 2

Reviewer 2 Report

The authors responded adequately to my questions and I think that the manuscript is improved substantially.